# SWAT: Sliding Window Adversarial Training for Gradual Domain Adaptation

## Abstract

Domain shifts are critical issues that harm the performance of machine learning. Unsupervised Domain Adaptation (UDA) mitigates this issue but suffers when the domain shifts are steep and drastic. Gradual Domain Adaptation (GDA) alleviates this problem in a mild way by gradually adapting from the source to the target domain using multiple intermediate domains. In this paper, we propose Sliding Window Adversarial Training (SWAT) for GDA. SWAT first formulates adversarial streams to connect the feature spaces of the source and target domains. Then, a sliding window paradigm is designed that moves along the adversarial stream to gradually narrow the small gap between adjacent intermediate domains. When the window moves to the end of the stream, i.e., the target domain, the domain shift is explicitly reduced. Extensive experiments on six GDA benchmarks demonstrate the significant effectiveness of SWAT, especially 6.1% improvement on Rotated MNIST and 4.1% advantage on CIFAR-100C over the previous methods.

## 1 Introduction

Traditional machine learning assumes identical training-test data distributions, yet real-world domain shifts often break this assumption and degrade model performance (Farahani et al., 2021). Unsupervised Domain Adaptation (UDA) is proposed to mitigate domain shifts by aligning feature distributions between a labeled source domain and an unlabeled target domain (Pan & Yang, 2009; Hoffman et al., 2018). Nevertheless, existing works (Kang et al., 2019; Tang & Jia, 2020; Yang et al., 2020) have revealed that when the domain gaps are large, directly aligning two domains not only fails to reduce the domain gaps, but even causes the negative transfer (Pan & Yang, 2009).

Gradual Domain Adaptation (GDA) (Kumar et al., 2020) is proposed to alleviate this problem in a mild way by gradually adapting from the source to the target domain using multiple intermediate domains, as shown in Fig. 1. This paper addresses the GDA problem through adversarial training. Adversarial training has been widely used in UDA and achieved impressive performance. This training paradigm, however, faces two challenges when applying in GDA. On the one hand, previous adversarial training methods (e.g., DANN (Ganin & Lempitsky, 2015b)) globally align two distributions through the game between generator and discriminator. This global matching cannot handle the continuous intermediate domains in GDA (Pei et al., 2018; Shi & Liu, 2024). As a result, the GDA problem degrades to the more difficult UDA problem.

On the other hand, the steep gradient of the adversarial training for large domain shifts will cause discontinuities and unsmooth problems in the manifold space (Rangwani et al., 2022; Shi & Liu, 2024; Zhang et al., 2019). As machine learning methods rely on the continuous and smooth manifold hypothesis to avoid abrupt changes in decision boundaries, this discontinuity and unsmoothness will cause error accumulation (Kumar et al., 2020; He et al., 2023; Xiao et al., 2024).

Towards the smooth and stable distribution matching, we propose sliding window mechanism for adversarial training. As a new training paradigm, the sliding window mechanism emerges three advantages over the traditional adversarial training: (1) *Locality*: The sliding window mechanism avoids global alignment by localizing the adversarial training range, i.e., it decomposes the continuous domain flow into multiple windows, and perform adversarial training in each window to gradually refine the alignment. The generator only focuses on the distribution differences in the current window, which reduces the complexity of the adversarial training. (2) *Dynamic*: The window is gradually shifted from the source domain to the target with the training process, and the update

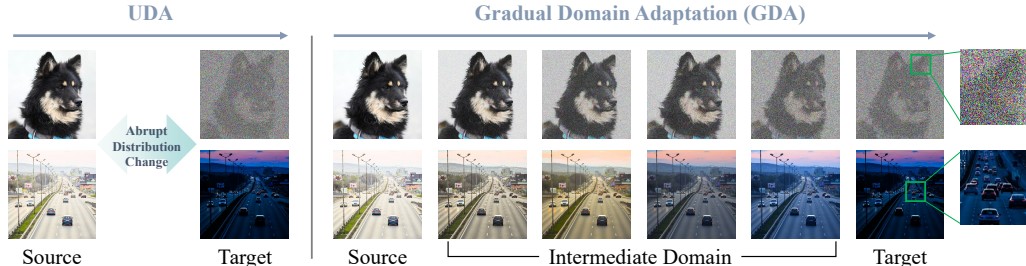

Figure 1: Comparison of UDA and GDA. Left (UDA): A single alignment maps source features directly onto the target domain. Right (GDA): Adaptation proceeds through a sequence of intermediate domains that smoothly adapt across domains, reducing abrupt distribution shifts.

frequency of generator parameters is synchronized with the speed of domain alignment, avoiding the error accumulation caused by large optimization step in traditional adversarial training. (3) *Continuity*: The sliding window continuously slides on the domain stream $H_z$, and the continuous change of parameter $p$ from 0 to 1 gradually turns the optimization focus from the left domain $H_l$ to the right domain $H_r$, avoiding the discrete switching in the multi-stage training.

Incorporated the sliding window mechanism, we present Sliding Window Adversarial Training (SWAT) method for GDA. Specifically, SWAT first formulates a bidirectional adversarial flow. This flow is optimized by a curriculum-guided sliding window, which finely controls the transition step between the source and the target domains, avoiding quadratic error accumulation caused by large transfer steps of the existing self-training strategy (Kumar et al., 2020). In the adversarial training phase, the flow generator enforces domain continuity through sliced Wasserstein optimization across evolving domains, while the discriminator progressively filters out source-specific features through adversarial training. This synergistic optimization achieves simultaneous domain invariance and target discriminability (Xiao et al., 2024). The contributions are summarized as follows:

1. We propose a sliding window mechanism to improve the adversarial training, which decomposes large domain shifts into multiple micro transfers through local, dynamic and continuous feature alignment, enabling stable and fine-grained distribution matching.

2. We present the Sliding Window Adversarial Training (SWAT) method for GDA. SWAT can adaptively align localized domain regions, mitigating error accumulation and enabling smooth and robust knowledge transfer.

3. Experiments on Rotated MNIST (96.7% vs. 90.6% SOTA), Portraits (87.4% vs. 86.16% SOTA) and CIFAR-100C (24.8% vs. 28.9%) demonstrate the effectiveness of SWAT.

## 2    RELATED WORK

**Unsupervised Domain Adaptation (UDA)** aims to mitigate domain shifts by aligning feature distributions between labeled source and unlabeled target domains. Traditional approaches leverage statistical measures like Maximum Mean Discrepancy (MMD) (Chen et al., 2020) to enforce domain invariance, but face limitations under severe distribution divergence: rigid MMD-based alignment risks distorting classifier boundaries by forcibly aligning non-overlapping supports (Zhao et al., 2019), while direct source-target alignment may erase discriminative structures, causing *negative transfer* (Tang & Jia, 2020; Yang et al., 2020). Adversarial methods like DANN (Ganin & Lempitsky, 2015a) and CDAN (Long et al., 2018) advanced alignment via adversarial training but enforce fixed pairwise alignment, leading to mode collapse under disjoint supports (Zhao et al., 2019) or gradient competition under large gaps (Pezeshki et al., 2021). While spectral regularization (Pezeshki et al., 2021) partially alleviates these issues, it retains rigid alignment steps.

**Gradual Domain Adaptation (GDA)** addresses scenarios where domain shifts occur incrementally, decomposing the overall distribution gap into smaller, more manageable steps through intermediate domains (Farshchian et al., 2018; Kumar et al., 2020). Existing methods employ diverse strategies to model these transitions: self-training leverages pseudo-labeling to bootstrap target predictions (Xie et al., 2020), gradient flow-based geodesic paths enforce smooth transitions via Rieman-

nian manifolds (Zhuang et al., 2024), style-transfer interpolation synthesizes intermediate domains through low-level feature mixing (Marsden et al., 2024), and optimal transport (OT) aligns domain distributions using Wasserstein distances (He et al., 2023). While alignment alone may rigidly match marginal distributions at the expense of discriminative structures. These issues are exacerbated in multi-step adaptation, where imperfectly aligned intermediates compound errors, leading to irreversible distortion of decision boundaries. Our SWAT framework uniquely preserves source-acquired information through bidirectional alignment, balancing between stability and plasticity,

**Adversarial Domain Adaptation** frameworks, including DANN (Ganin & Lempitsky, 2015a) and CDAN (Long et al., 2018), aligh the source and target domains through adversarial training. These methods employ gradient reversal layers or conditional adversarial networks to learn domain-invariant representations. However, these methods enforce fixed pairwise alignment between source and target domains, leading to mode collapse when domain supports are disjoint (Zhao et al., 2019) or under large distribution gaps due to gradient competition (Pezeshki et al., 2021). Recent advances, such as spectral regularization (Pezeshki et al., 2021), partially alleviate these issues but retain the rigidity of discrete alignment steps. In contrast, SWAT reformulates domain adaptation as a *continuous manifold transport process*. By constructing intermediate domains along a feature transport flow, SWAT avoids abrupt transitions and assimilates novel target modes progressively, i.e., a critical failure point for conventional UDA and adversarial methods.

## 3 PROBLEM SETUP

**Domain Space**  Let $\mathcal{X} \subseteq \mathbb{R}^d$ denote the input space and $\mathcal{Y} = \{1, ..., k\}$ the label space. We model each domain as a joint probability distribution $P_t(X, Y) = P_t(X)P_t(Y|X)$ over $\mathcal{Z} = \mathcal{X} \times \mathcal{Y}$, where $t \in \{0, ..., n\}$ indexes domains along the adaptation path.

**Gradually Shifting Domain**  In the gradually domain setting(Kumar et al., 2020), given a sequence of domains $\{P_t\}_{t=0}^n$ with gradually shifting distributions, where $P_0$ is the labeled source domain and $P_n$ the unlabeled target domain, GDA aims to learn a hypothesis $h : \mathcal{X} \to \mathcal{Y}$ that minimizes target risk $\epsilon_n(h)$, under two core assumptions (Kumar et al., 2020; Long et al., 2015): (1) the distribution shifts between consecutive domains are limited, known as bounded successive divergence, and (2) the conditional distribution of labels given inputs remains unchanged across domains, referred to as conditional invariance:

$$\mathcal{W}_1(P_t, P_{t+1}) \leq \Delta, \quad \mathcal{P}_t(Y|X) = \mathcal{P}_{t+1}(Y|X), \quad \forall t \in \{0, ..., n-1\}, \tag{1}$$

where $\mathcal{W}_1$ is the Wasserstein-1 distance and $\Delta$ quantifies maximum inter-domain drift. Conditional probability consistency ensures that label semantics remain stable during adaptation.

**Model Pretraining in the source domain**  The goal of pretraining in the source domain is to learn a model $C : \mathcal{X} \to \mathcal{Y}$ that maps input features $x$ from the training data set $\mathcal{D} = \{(x, y)\}$ to their corresponding labels $y$. Considering the loss function $l$, the classifier optimized on $\mathcal{D}_t$ is denoted by $C$, defined as:

$$C = \arg \min_C \mathbb{E}_{(x,y) \sim \mathcal{D}_t}[l(C(x), y)]. \tag{2}$$

**Gradual Domain Adaptation**  Gradual domain adaptation aims to train a model $C$ that effectively generalizes to the target domain $\mathcal{D}_n$ by incrementally transferring knowledge from the labeled source domain $\mathcal{D}_0$ through a sequence of unlabeled intermediate domains $\mathcal{D}_1, \mathcal{D}_2, \ldots, \mathcal{D}_{n-1}$. The adaptation process involves multi-step pseudo-labeling and self-training, where the model $C_0$ is trained on the source domain and then adapted to the intermediate domains by the following self-training procedure $\text{ST}(C_t, \mathcal{D}_t)$:

$$\text{ST}(C_t, \mathcal{D}_t) = \arg \min_{C'} \mathbb{E}_{x \sim \mathcal{D}_t}[l(C'(x), \hat{y}_t(x))]. \tag{3}$$

In particular, $\hat{y}_t(x) = \text{sign}(C_t(x))$ is the pseudo-label generated by the model $C_t$ for unlabeled data of $\mathcal{D}_t$, where $\mathcal{D}_t$ denotes the unlabeled intermediate domain. Meanwhile, $C'$ is the next learned model, also denoted by $C_{t+1}$.

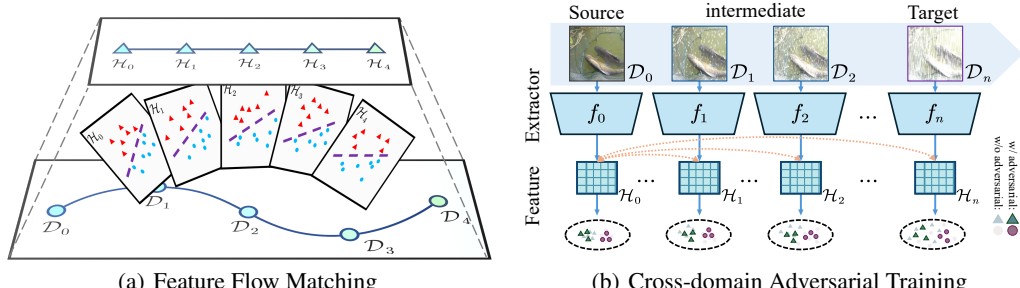

(a) Feature Flow Matching          (b) Cross-domain Adversarial Training

Figure 2: (a) Illustration of sliding window mechanism, where overlapping feature spaces facilitate smooth domain transitions along the domain sequence. (b) Incremental domain alignment with the SWAT framework using adversarial training to preserve task performance and encourage feature consistency across domains.

## 4 METHODOLOGY

The proposed SWAT decomposes large domain shifts into manageable local transitions, enabling stable and precise distribution matching for GDA. The core components are detailed in the following.

**Continuous Feature Flow**   As illustrated in Fig. 2(a), SWAT defines a continuous sequence of feature distributions:

$$\{\mathcal{H}_t\}_{t\in[0,n]}, \quad \mathcal{H}_t = p_t(\mathbf{h}), \tag{4}$$

over the latent space $\mathcal{H} \subseteq \mathbb{R}^z$, where each $\mathcal{H}^i$ is the feature manifold at adaptation step $i$. Here $f_t : \mathcal{X} \to \mathcal{H}$ and $g_t : \mathcal{H} \to \mathcal{Y}$ denote the encoder and classifier at step $t$. The overall model $g_t \circ f_t$ smoothly evolves from $(f_0, g_0)$ on the source domain to $(f_n, g_n)$ on the target.

Existing GDA methods formulate each pair $(f_i, g_i)$ as an independent stage in the adaptation process, where $(f_{i+1}, g_{i+1})$ is trained after $(f_i, g_i)$ has converged. This leads to a sequence of discrete transitions between domains. In contrast, SWAT learns a continuous sequence of models $\{(f^z, g^z)\}_{z\in[0,n]}$ and aligns features along the entire path $\mathcal{H}^z$. This continuous flow matching avoids abrupt transitions and enables fine-grained adaptation at every intermediate point.

Unlike the discrete adaptation process in previous GDA methods, SWAT enables continuous feature transferring along the domain stream $\mathcal{H}^z$ ($z \in [0, n]$) through a sliding window, as illustrated in Fig. 2(b).

**Sliding Window Mechanism**   At any step $l \in \{0, \dots, n-1\}$, *sliding window* is the pair of adjacent domains $\{\mathcal{H}_l, \mathcal{H}_r\}$ with $r = l+1$. The scalar parameter $p \in [0, 1]$ controls where within that window we align:

$$\mathcal{H}^{(l+p)} = (1-p)\,\mathcal{H}_l + p\,\mathcal{H}_r, \tag{5}$$

where $\mathcal{H}^{(l+p)}$ refers to a domain located between $\mathcal{H}_l$ and $\mathcal{H}_r$. Here, $\mathcal{H}_l$ and $\mathcal{H}_r$ refer to the left and right critical domains, respectively. When $p$ reaches 1, the window "slides" one step to the right (i.e. $l \leftarrow l + 1$ and $p$ resets toward 0), hence the next window will be $\{\mathcal{H}_{l+1}, \mathcal{H}_{l+2}\}$. As $p$ varies from 0 to 1 and then triggers a slide, SWAT walks continuously through the entire domain stream. We then formalize the sliding-path alignment as:

$$\mathcal{H}_0 \leftrightarrow \mathcal{H}^{(l+p)}, \quad l \in \{0, 1, \dots, n-1\}, \quad p \in [0, 1], \tag{6}$$

where both $l$ and $p$ are parameters controlling smooth transitions across domains. This formulation enables fine-grained domain alignment through continuously shifting intermediate representations.

**Bidirectional Flow Matching**   Building upon the sliding window mechanism, we further incorporate it with the adversarial training for smooth flow matching. Specifically, we define $G_m$ as the transformation function that maps a representation $h$ from the source domain space $\mathcal{H}_s$ to a target domain within the domain stream $\mathcal{H}^z$, where $z \in [0, n]$ indicates the position of the target domain within the stream. Conversely, $G_s$ denotes the reverse transformation, mapping features from any

domain in the stream $\mathcal{H}^z$ back to the source domain $\mathcal{H}_s$. Thus, SWAT can be expressed as the bidirectional transformations:

$$G_m : \mathcal{H}_s \to \mathcal{H}^z, \quad G_s : \mathcal{H}^z \to \mathcal{H}_s. \tag{7}$$

We employ the Wasserstein GAN (WGAN) (Arjovsky et al., 2017) to train the SWAT model, as the Wasserstein distance provides a more effective measure of the distance between domains, generates higher-quality target domains $\mathcal{H}^z$, and is easier to train. The objective function for the adversarial training module is defined as:

$$\min_D \max_G V(\mathbb{P}_g, \mathbb{P}_r) = \min_D \max_G \mathbb{E}_{\substack{\hat{h} \sim \mathbb{P}_g \\ h \sim \mathbb{P}_r}} \left[ D(\hat{h}) - D(h) \right] + \mathcal{R}, \tag{8}$$

where $\hat{h}$ represents a representation generated by the generator $G$, which approximates the target domain distribution $\mathbb{P}_g$. $h$ is a representation from the real data distribution $\mathbb{P}_r$, corresponding to actual data from the target domain. $D$ denotes the discriminator of the corresponding domain, and different domains have different discriminators. $\mathcal{R}$ represents the regularization term proposed by Gulrajani et al. (2017):

$$\mathcal{R} = \mathbb{E}_{\tilde{h} \sim \mathbb{P}_{\tilde{h}}} \left[ \lambda \left( ||\nabla_{\tilde{h}} D(\tilde{h})||_2 - 1 \right)^2 \right], \tag{9}$$

where $\tilde{h}$ denotes a random linear interpolation of points from $\hat{h}$ and $h$ representations, and $\lambda$ is a hyperparameter controlling the strength of the regularization.

To facilitate bidirectional feature alignment between the source domain $\mathcal{H}_0$ and the critical domains, we formulate bidirectional flow matching based on the minimax objective $V(\mathbb{P}_g, \mathbb{P}_r)$ defined in Eq. (8). Without loss of generality, taking the left critical domain $H_l$ as an example, the adversarial loss enforces cross-domain distribution matching through dual mapping paths:

$$\mathcal{L}^l_{\text{adv}} = V(G_m(\mathcal{H}_0), \mathcal{H}_l) + V(G_s(\mathcal{H}_l), \mathcal{H}_0), \tag{10}$$

where $G_m$ maps source features to the critical domain while $G_s$ reconstructs the original domain. The symmetrical adversarial loss $\mathcal{L}^r_{\text{adv}}$ for the right critical domain $\mathcal{H}_r$ follows the same dual-path formulation.

**Semantic Consistency Preservation**  To prevent mode collapse and maintain content integrity during adaptation, we employ cycle-consistent regularization inspired by CycleGAN (Zhu et al., 2017). This ensures that features cyclically transformed through $\mathcal{H}_0 \to \mathcal{H}_l \to \mathcal{H}_0$ to preserve semantic consistency:

$$\mathcal{L}^l_{\text{cycle}} = \mathbb{E}_{h \sim \mathcal{H}_0} \left[ ||G_s(G_m(h)) - h||_2 \right] + \mathbb{E}_{h \sim \mathcal{H}_l} \left[ ||G_m(G_s(h)) - h||_2 \right]. \tag{11}$$

The bidirectional reconstruction regularizations enforce invertible transformations while penalizing semantic distortions, particularly crucial for preserving task-relevant features in critical domains.

### 4.1 THE OVERALL OBJECTIVE

Following previous GDA methods, we optimize the self-training loss as follows:

$$\mathcal{L}^l_{st} = \mathbb{E}_{h \sim \mathcal{H}}[l(g(h), \hat{y}(h))], \tag{12}$$

where $l$ is the cross-entropy loss. When $h$ is from the un-labeled domain, $\hat{y}_t(x)$ is the pseudo-label generated by the model $g$. When $h$ is a feature generated by $G_m(h_0)$, it represents the ground-truth label of the original representation $h_0$ from the source domain.

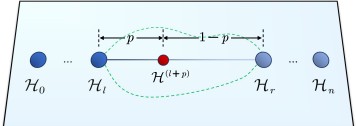

As illustrated in Fig.3, the generated feature space $\mathcal{H}^{(l+p)}$ is enforced to satisfy the condition $\text{dist}(\mathcal{H}_l, \mathcal{H}^{(l+p)})/\text{dist}(\mathcal{H}_r, \mathcal{H}^{(l+p)}) = p/(1-p)$, where $\text{dist}(\cdot, \cdot)$ denotes a valid distance metric between two distributions (Peyré et al., 2019; Arjovsky et al., 2017).

Figure 3: $\mathcal{H}^{(l+p)}$ lies along the smooth and continuous domain flow determined by $p$.

The overall objective is formulated as follows:

$$\mathcal{L} = (1 - p)\mathcal{L}^l + p\mathcal{L}^r, \tag{13}$$

where $\mathcal{L}^l$ is the adversarial training loss defined as: $\mathcal{L}^l = \mathcal{L}^l_{adv} + \mathcal{L}^l_{cycle} + \mathcal{L}^l_{st}$. By optimizing Eq. 13, we achieve continuous flow matching in the feature space. For clear understanding, we summarize the main idea of SWAT in Algorithm 1.

---

**Algorithm 1:** Sliding Window Adversarial Training (SWAT)

---

**Input:** Domains $\mathcal{D}_0$ (source), $\mathcal{D}_1, \ldots, \mathcal{D}_n$ (target); pretrained encoder $f$ and classifier $g$.
**Output:** $\frac{1}{N}\sum_{i=1}^{N}\mathbf{1}(\hat{y}_i = \mathcal{Y}_i)$ where $\hat{y} = g(f(\mathcal{D}_n))$.
Initialize generators $G_m, G_s$ and discriminators $D_s, D_l, D_r$.
**for** $l \leftarrow 0$; $l < n$; $l \leftarrow l + 1$ **do**
    $r \leftarrow l + 1$
    $D_l \leftarrow D_r$; reinitialize $D_r$.
    $\mathcal{H}_0 \leftarrow f(\mathcal{D}_0), \mathcal{H}_l \leftarrow f(\mathcal{D}_l), \mathcal{H}_r \leftarrow f(\mathcal{D}_r)$
    **for** $p \leftarrow 0$; $p \leq 1$; $p \leftarrow p + \Delta p$ **do**
        $\mathcal{L} \leftarrow (1 - p)\, L(G_m, G_s, D_s, D_l, \mathcal{H}_0, \mathcal{H}_l) \ + \ p\, L(G_m, G_s, D_s, D_r, \mathcal{H}_0, \mathcal{H}_r)$
        update $G_m, G_s, D_s, D_l, D_r, f, g$           // e.g., Adam
    **end**
**end**

---

## 5 EXPERIMENTS

### 5.1 DATASETS AND IMPLEMENTATION DETAILS

Following the standard GDA protocol, we conduct extensive experiments on 6 datasets. **Rotated MNIST** is constructed from MNIST (Deng, 2012), this dataset contains 50,000 source domain images (original digits) and 50,000 target domain images rotated by 45°. Intermediate domains interpolate rotation angles between 0° and 45°. **Color-Shift MNIST** images are normalized to [0,1] for the source domain and shifted to [1,2] for the target domain (He et al., 2023), with intermediate domains generated by linearly interpolating color intensity. **Portraits** (Ginosar et al., 2015) are chronologically divided into 9 temporal domains (1905–2013), each with 2,000 images (Kumar et al., 2020). The first and last domains serve as source/target; images are resized to 32×32 pixels. **Cover Type** (Blackard, 1998) tabular dataset sorted by horizontal distance to water, uses 50,000 source samples, 10×40,000 intermediate domains, and 50,000 target samples (Kumar et al., 2020) for classifying spruce fir vs. Rocky Mountain pine.

To evaluate the performance of GDA methods under high-severity shifts, we introduce a new evaluation protocol using the corruption benchmarks **CIFAR-10C** and **CIFAR-100C** (Hendrycks & Dietterich, 2019). Each benchmark applies 15 corruption types at 5 severity levels to the validation and test splits of CIFAR (Krizhevsky et al., 2009). We regard the clean training images as the source domain, the images of severity levels 1–4 (across all corruption types) as a sequence of intermediate domains, and treat severity level 5 as the target domain. Following the RobustBench benchmark (Croce et al., 2020; Croce & Hein, 2020), WideResNet-28 (Zagoruyko & Komodakis, 2016) and ResNeXt-29 (Xie et al., 2017) used as the source model for CIFAR10-to-CIFAR10C and CIFAR100-to-CIFAR100C, respectively.

All results are averaged over 5 runs. Please refer to section A.1 for more detailed implementation.

### 5.2 EXPERIMENTAL RESULTS

Table1 reports the experiment results on Rotated MNIST, Color-Shift MNIST, Portraits and Cover Type, respectively.

Compared with the UDA methods (He et al., 2023), Table 1 highlights the clear advantage of GDA over traditional UDA. SWAT achieves the best accuracy on all benchmarks, indicating superior

Table 1: Comparison of domain adaptation methods on 4 GDA datasets.

| Methods | Gradual | Rotated MNIST | Color-Shift MNIST | Portraits | Cover Type |
|---|---|---|---|---|---|
| DANN (Ganin et al., 2016) | ✗ | 44.2 | 56.5 | 73.8 | - |
| DeepCoral (Sun & Saenko, 2016) | ✗ | 49.6 | 63.5 | 71.9 | - |
| DeepJDOT (Damodaran et al., 2018) | ✗ | 51.6 | 65.8 | 72.5 | - |
| GST (Kumar et al., 2020) (ICML'20) | ✓ | 83.8 | 74.0 | 82.6 | 73.5 |
| IDOL (Chen & Chao, 2021) (NeurIPS'21) | ✓ | 87.5 | - | 85.5 | - |
| AGST (Zhou et al., 2022) (IEEE'22) | ✓ | 76.2 | - | 77.6 | |
| GGF (Zhuang et al., 2024) (ICLR'24) | ✓ | 67.7 | - | 86.2 | |
| GOAT (He et al., 2023) (JMLR'24) | ✓ | 86.4 | 91.8 | 83.6 | 69.9 |
| DRO (Najafi et al., 2024) (NeurIPS'24) | ✓ | 53.2 | - | - | - |
| AST (Shi & Liu, 2024) (NeurIPS'24) | ✓ | 90.6 | - | 84.8 | - |
| CNF (Sagawa & Hino, 2025) (Neural Computation'25) | ✓ | 62.6 | - | 84.6 | - |
| SWAT (Ours) | ✓ | **96.7** | **99.6** | **87.4** | **75.0** |

Table 2: Comparison of classification error rates (%) at severity level 5 for TTA and GDA methods on CIFAR-10C and CIFAR-100C. Lower is better.

| | Method | Gradual | gaussian | shot | impulse | defocus | glass | motion | zoom | snow | frost | fog | brightness | contrast | elastic | pixelate | jpeg | Mean |
|---|---|---|---|---|---|---|---|---|---|---|---|---|---|---|---|---|---|---|
| CIFAR-10C | Source only | ✗ | 72.3 | 65.7 | 72.9 | 46.9 | 54.3 | 34.8 | 42.0 | 25.1 | 41.3 | 26.0 | 9.3 | 46.7 | 26.6 | 58.5 | 30.3 | 43.5 |
| | BN-1 | ✗ | 28.1 | 26.1 | 36.3 | 12.8 | 35.3 | 14.2 | 12.1 | 17.3 | 17.4 | 15.3 | 8.4 | 12.6 | 23.8 | 19.7 | 27.3 | 20.4 |
| | TENT-cont. (Wang et al., 2020) | ✗ | 24.8 | 20.6 | 28.6 | 14.4 | 31.1 | 16.5 | 14.1 | 19.1 | 18.6 | 18.6 | 12.2 | 20.3 | 25.7 | 20.8 | 24.9 | 20.7 |
| | AdaContrast (Chen et al., 2022) | ✗ | 29.1 | 22.5 | 30.0 | 14.0 | 32.7 | 14.1 | 12.0 | 16.6 | 14.9 | 14.4 | 8.1 | 10.0 | 21.9 | 17.7 | 20.0 | 18.5 |
| | CoTTA (Wang et al., 2022) | ✗ | 24.3 | 21.3 | 26.6 | 11.6 | 27.6 | 12.2 | 10.3 | 14.8 | 14.1 | 12.4 | 7.5 | 10.6 | **18.3** | 13.4 | **17.3** | 16.2 |
| | GTTA-MIX (Marsden et al., 2022) | ✗ | 23.4 | **18.3** | **25.5** | 10.1 | **27.3** | 11.6 | 10.1 | 14.1 | **13.0** | **10.9** | 7.4 | 9.0 | 19.4 | 14.5 | 19.8 | 15.6 |
| | GST (Kumar et al., 2020) | ✓ | 50.0 | 43.9 | 50.3 | 20.6 | 51.2 | 17.2 | 16.7 | 17.5 | 24.3 | 17.5 | 6.9 | 13.2 | 24.9 | 39.9 | 26.6 | 28.1 |
| | GOAT (He et al., 2023) | ✓ | 72.7 | 65.7 | 73.0 | 46.7 | 54.5 | 34.3 | 41.5 | 24.9 | 41.0 | 26.0 | 9.3 | 46.6 | 26.4 | 58.1 | 30.2 | 43.4 |
| | SWAT (ours) | ✓ | **21.4** | 20.0 | 26.8 | **9.7** | 28.5 | **10.2** | **8.4** | **3.1** | 13.4 | 11.5 | **7.1** | **8.9** | 19.8 | **13.3** | 20.1 | **15.4** |
| CIFAR-100C | Source only | ✗ | 73.0 | 68.0 | 39.4 | 29.3 | 54.1 | 30.8 | 28.8 | 39.5 | 45.8 | 50.3 | 29.5 | 55.1 | 37.2 | 74.7 | 41.2 | 46.4 |
| | BN-1 | ✗ | 42.1 | 40.7 | 42.7 | 27.6 | 41.9 | 29.7 | 27.9 | 34.9 | 35.0 | 41.5 | 26.5 | 30.3 | 35.7 | 32.9 | 41.2 | 35.4 |
| | TENT-cont. (Wang et al., 2020) | ✗ | 37.2 | 35.8 | 41.7 | 37.9 | 51.2 | 48.3 | 48.5 | 58.4 | 63.7 | 71.1 | 70.4 | 82.3 | 88.0 | 88.5 | 90.4 | 60.9 |
| | AdaContrast (Chen et al., 2022) | ✗ | 42.3 | 36.8 | 38.6 | 27.7 | 40.1 | 29.1 | 27.5 | 32.9 | 30.7 | 38.2 | 25.9 | 28.3 | 33.9 | 33.3 | 36.2 | 33.4 |
| | CoTTA (Wang et al., 2022) | ✗ | 40.1 | 37.7 | 39.7 | 26.9 | 38.0 | 27.9 | 26.4 | 32.8 | 31.8 | 40.3 | 24.7 | 26.9 | 32.5 | 28.3 | 33.5 | 32.5 |
| | GTTA-MIX (Marsden et al., 2022) | ✗ | 36.4 | 32.1 | 34.0 | 24.4 | 35.2 | 25.9 | 23.9 | 28.9 | 27.5 | 30.9 | 22.6 | 23.4 | 29.4 | 25.5 | 33.3 | 28.9 |
| | GST (Kumar et al., 2020) | ✓ | 49.8 | 56.7 | 32.3 | 22.5 | 41.6 | 25.0 | 23.3 | 30.3 | 32.2 | 38.1 | 22.1 | 27.0 | 33.1 | 40.8 | 35.8 | 33.3 |
| | GOAT (He et al., 2023) | ✓ | 73.4 | 67.9 | 39.1 | 28.7 | 53.8 | 30.2 | 28.7 | 39.3 | 45.7 | 50.0 | 29.4 | 53.7 | 36.8 | 74.3 | 41.2 | 46.2 |
| | SWAT (ours) | ✓ | **28.6** | **26.9** | **23.5** | **22.3** | **29.0** | **22.7** | **22.4** | **24.4** | **24.3** | **25.7** | **21.5** | **22.7** | **26.5** | **23.4** | **28.8** | **24.8** |

representation transfer performance under gradual shifts. Per-dataset trends across different numbers of given domains (2–6) further corroborate this advantage, especially when only a few domains are available, as illustrated in Appendix A.2, Table 5. Additional experiments and computational-cost comparisons are provided in sections A.3 and B.

Table 2 presents classification error rates (severity level 5) on CIFAR-10C and CIFAR-100C. We group methods into two families: Test-Time Adaptation (TTA) and GDA. For TTA, "Source only" refers to the fixed pretrained model, and BN-1 updates batch normalization statistics on each test batch. The other baselines, including TENT-continual (Wang et al., 2020), AdaContrast (Chen et al., 2022), CoTTA (Wang et al., 2022), and GTTA-MIX (Marsden et al., 2022), perform online adaptation of either the feature extractor or the classifier. Our approach, SWAT, combines the stability of batch-norm re-estimation with sample-wise alignment. Across both benchmarks, SWAT achieves the lowest mean error (15.4% on CIFAR-10C, 24.8% on CIFAR-100C), outperforming the strongest prior TTA (GTTA-MIX: 15.6%/28.9%) and GDA competitors on nearly every corruption type.

## 5.3 DOMAIN SHIFTS ANALYSIS

**Quantitative Analysis of Domain Shifts** We employ $\mathcal{A}$-distance (Ben-David et al., 2010) as the proxy of $\mathcal{H}\Delta\mathcal{H}$ distance to quantitatively evaluate the domain shifts, as illustrated in Fig. 4. We observe that the $\mathcal{A}$-distance between the source domain $\mathcal{H}_0$ and the target domain $\mathcal{H}_n$ exhibits large fluctuations (peak at 1.498), which indicates that directly aligning two domains causes unstable transfer or even negative transfer when the domain shifts are significantly large. In contrast, the proposed SWAT maintains near-zero distances ($< 0.11$) to critical intermediate domains $\mathcal{H}_l, \mathcal{H}_r$ across all positions, achieving a 63.7% reduction in the average $\mathcal{A}$-distance between $\mathcal{H}_0$ and $\mathcal{H}_n$ (0.104

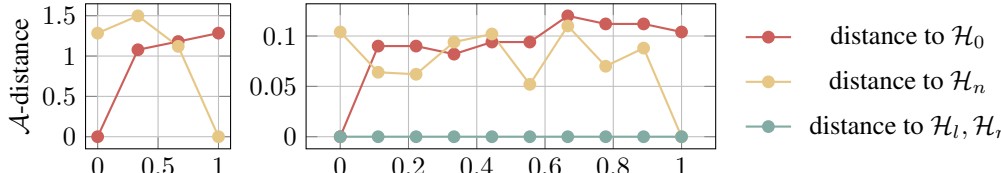

Figure 4: The figure illustrates how the interpolated domains evolve in domain discrepancy along the adaptation path. The horizontal axis denotes the interpolation place $z \in [0, 1]$ (0 = source domain, 1 = target domain). The vertical axis represents the $\mathcal{A}$-distance (Ben-David et al., 2010; Mansour et al., 2009), a proxy for distribution divergence. On the left, the $\mathcal{A}$-distance is computed with representations of a fixed encoder, while on the right, the distance is calculated using our SWAT representation.

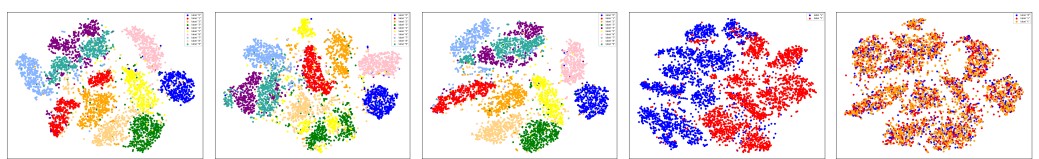

(a) Source space $\mathcal{H}_0$   (b) Target space $\mathcal{H}_n$ w/o flow matching   (c) Target space $\mathcal{H}_n$ w/ flow matching   (d) $\mathcal{H}_0, \mathcal{H}_n$ w/o adversarial training   (e) $\mathcal{H}_0, \mathcal{H}_g, \mathcal{H}_n$ w/ adversarial training

Figure 5: t-SNE visualization of feature space geometry under different domain adaptation strategies through two complementary perspectives on Rotated MNIST (with 4 intermediate domains).

vs. 1.284), demonstrating smooth knowledge transfer. The symmetrical reduction of bidirectional distances confirms balanced adaptation between forward and backward domain transitions.

**Visualization Analysis of Domain Shifts**   The t-SNE visualizations (Fig. 5) reveal the geometric impact of different strategies: (1) Direct mapping to $\mathcal{H}_n$ without flow matching (Fig. 5(b)) causes catastrophic cluster overlap, as rigid alignment disrupts local semantic structures. (2) SWAT (Fig. 5(c)) maintains a high percentage of $\mathcal{H}_0$'s cluster purity through flow matching that preserves isometric relationships between neighboring domains $\mathcal{H}_l \leftrightarrow \mathcal{H}_r$. (3) The non-adversarial path $\mathcal{H}_0 \rightarrow \mathcal{H}_n$ (Fig. 5(d)) exhibits discontinuous jumps (Hausdorff distance 4.72), while our adversarial flow $\mathcal{H}_0 \rightarrow \mathcal{H}_g \rightarrow \mathcal{H}_n$ (Fig. 5(e)) reduces trajectory fragmentation by 75.6% (Hausdorff 1.15). This geometric perspective demonstrates the Semantic invariance and topological continuity of SWAT in the feature space.

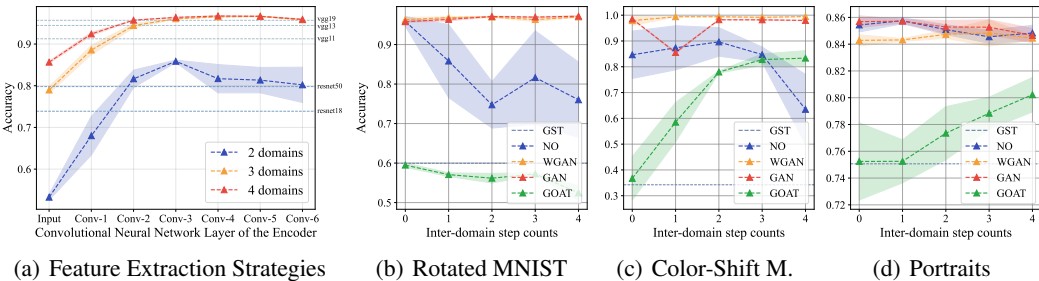

(a) Feature Extraction Strategies   (b) Rotated MNIST   (c) Color-Shift M.   (d) Portraits

Figure 6: Ablation analysis of SWAT. (a) Comparison of flow matching with feature extraction strategies. (b-d) Performance of SWAT on Rotated MNIST, Color-Shift MNIST, and Portrait with 3 intermediate domains, showing the accuracy changes of different training strategies (NO: no adversarial, GST, GOAT with 0-4 inter-domain step counts.

## 5.4   ABLATION STUDY

**Continuous Feature Flow**   By progressively enabling multi-scale feature aggregation in our sliding window framework, we observe significant performance improvements across 2–4 domain set-

Table 3: Trends analysis of $p$. "Ours" denotes gradually increasing $p$ at equal intervals, "Fixed" keeps $p$ a constant value of 0.5, "Rand" samples $p$ randomly from a uniform distribution $U(0, 1)$ at each step, and "Sorted" adopts a fixed set of random values in ascending order.

| Methods | Rotated MNIST | | | | | Portraits | | | | |
|---|---|---|---|---|---|---|---|---|---|---|
| | 0 | 1 | 2 | 3 | 4 | 0 | 1 | 2 | 3 | 4 |
| Ours | 83.3±0.9 | 85.0±0.5 | **86.1**±0.4 | **86.9**±0.2 | **88.1**±1.5 | 82.9±1.2 | **84.6**±0.2 | **85.0**±0.9 | **85.1**±0.2 | **85.3**±0.1 |
| Sorted | **84.1**±0.8 | **86.4**±0.6 | 86.0±1.7 | 86.3±0.1 | 85.7±0.5 | 82.7±0.5 | 84.0±0.6 | 84.2±0.1 | 84.3±0.2 | 84.5±0.1 |
| Rand | 83.4±0.2 | 80.9±7.0 | 84.5±2.8 | 86.3±0.9 | 86.1±0.4 | 82.4±0.5 | 84.0±0.6 | 84.2±0.2 | 84.3±0.2 | 84.1±0.1 |
| Fixed | 83.3±0.0 | 83.7±0.0 | 83.8±0.1 | 83.8±0.1 | 84.1±0.0 | **83.9**±0.3 | 83.4±0.0 | 83.1±2.1 | 84.1±0.1 | 84.7±0.3 |

Table 4: Ablation study on CIFAR-10C. The error rates (%) are computed across 6 specified domains.

| Setup | gaussian | shot | impulse | defocus | glass | motion | zoom | snow | frost | fog | brightness | contrast | elastic | pixelate | jpeg | Mean |
|---|---|---|---|---|---|---|---|---|---|---|---|---|---|---|---|---|
| The full model | **21.4** | **20.0** | **26.8** | **9.7** | **28.5** | **10.2** | **8.4** | **3.1** | **13.4** | **11.5** | **7.1** | **8.9** | **19.8** | **13.3** | 20.1 | **15.4** |
| w/o $\mathcal{L}_{adv}$ | 26.9 | 45.2 | 37.6 | 47.9 | 39.3 | 18.7 | 34.7 | 16.0 | 31.0 | 61.1 | 60.6 | 58.4 | 58.5 | 48.7 | **19.8** | 40.3 |
| w/o $\mathcal{L}_{cycle}$ | 30.3 | 40.0 | 46.3 | 29.1 | 35.4 | 26.1 | 57.1 | 21.8 | 18.2 | 46.3 | 98.8 | 36.4 | 62.0 | 53.9 | 21.9 | 41.6 |
| w/o $\mathcal{L}_{st}$ | 27.8 | 26.1 | 35.8 | 13.6 | 34.8 | 13.8 | 12.0 | 16.9 | 17.5 | 15.5 | 7.8 | 12.2 | 23.4 | 20.8 | 27.2 | 20.3 |

tings for Rotated MNIST (Fig. 6(a)). The shallowest setting, corresponding to a shallow neural network without adversarial training, performs over 25% worse than our feature flow matching approach, highlighting the limitations of low-level features in capturing transferable representations.

**Bidirectional Flow Matching**  In Rotated MNIST dataset (Fig. 6(b)), the accuracy without any adversarial alignment (NO) drops significantly with inter-domain steps, whereas incorporating SWAT with feature flow matching improves accuracy. For the Color-Shift MNIST dataset (Fig. 6(c)), SWAT significantly enhances accuracy, achieving near-optimal performance across inter-domain steps. In the Portraits dataset (Fig. 6(d)), SWAT outperforms the baseline NO method and any previous static transport methods.

**Sliding Window Mechanism**  Table 3 reports results with different inter-domain adaptation steps. The sliding window mechanism (Ours) consistently achieves the best average accuracy, e.g., 88.1% vs. 84.1% (Fixed) and 86.1% (Rand) on Rotated MNIST with substantially lower standard deviation (±0.5 vs. ±7.0 at step 1). These results confirm that gradually increasing $p$ produces more stable and optimal adaptation than holding $p$ fixed, sampling it at random, or reordering random draws.

**Effectiveness of Different Components**  In Table 4, removing any of the three components degrades robustness, but the effects are asymmetric. The three components are complementary: adversarial alignment reduces shift, cycle-consistency regularizes the adaptation trajectory, and self-training refines supervision, which is consistent with our broader ablations as shown in Table 4, showing that removing alignment or label-quality mechanisms significantly degrades performance.

## 6  CONCLUSION

This work proposes a sliding window mechanism to improve the adversarial training, which splits large domain shifts into multiple micro transfers through local, dynamic and continuous feature alignment, enabling fine-grained distribution matching. Building upon this training paradigm, we present the Sliding Window Adversarial Training, a novel framework for GDA that incorporates the sliding window mechanism with adversarial flow matching to enable continuous and stable feature alignment. Extensive experimental results demonstrate the superior effectiveness and robustness of SWAT across diverse benchmarks.

## REPRODUCIBILITY STATEMENT

The problem setup, algorithm, and notation are specified in Secs. 3–4 (incl. Algorithm 1 and Figs. 2–3), while dataset construction and implementation details, including architectures, optimizers, hyperparameters, and hardware, are documented in Sec. 5.1 and App. A.1. Our evaluation protocol and metrics are described in Secs. 5.2–5.3, and multiple-run reporting (five seeds) with variability is summarized in Sec. 5.2 and expanded in App. A.2–A.3 (Tables 5–9). Upon acceptance, we will open-source the full source code, including training/evaluation scripts, configuration files, and pretrained checkpoints to exactly reproduce all tables and figures.

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

## A    EXPERIMENTAL DETAILS

### A.1    IMPLEMENTATION

For the Rotated MNIST, Color-Shift MNIST, and Portraits datasets, we implemented a CNN with three convolutional layers with 32 channels. After the encoder, we added a fully connected classifier with two hidden layers of 256 units each. For the Cover Type dataset, we adopted a similar approach using three fully connected layers with ReLU activations, where the hidden dimensions increase from 128 to 256 to 512 units, ending with an output layer matching the number of classes.

Our transport architecture includes generators composed of a single residual block containing three linear layers. The discriminator is built with three linear layers, each having 128 hidden units and paired with ReLU activation functions. We used the Adam optimizer for optimization (Kingma & Ba, 2014), Dropout for regularization (Srivastava et al., 2014), and Batch Normalization to stabilize training (Ioffe & Szegedy, 2015). The number of intermediate domains generated between source and target domains is treated as a hyperparameter, with the model's performance evaluated for 0, 1, 2, 3, or 4 intermediate domains. All the code was run on NVIDIA RTX 4090 GPUs.

In addition, we followed (Kumar et al., 2020) to filter out the 10% of data points where the model's predictions exhibit the least confidence. However, instead of relying on the typical uncertainty measure, we define the confidence level as the difference between the largest and the second-largest values in the model's output. We have found that this produces better results and we use this setting in all comparative tests.

We pretrain the encoder and classifiers $f, g$ on four datasets, and the results of the pretrain are shown in Fig. 7, where the accuracy varies across multiple domains. All of our experiments, including ablations on the GOAT, GST method in section 5.4, are performed using the same pretrained model. With a total of six domains in the setup, the precision of the four datasets for the classifications trained on the source domain directly using the classification results in the subsequent domains are shown in Fig. 7. The accuracies fall roughly stepwise in line with our expectations for the problem setup.

### A.2    COMPARATIVE EXPERIMENT

Table 5 compares SWAT against GST (Kumar et al., 2020) and GOAT (He et al., 2023) on both vision benchmarks and a tabular dataset (Cover Type), using the same encoder–classifier architecture and low-confidence sample selection strategy throughout. SWAT consistently outperforms GST and GOAT across every setting, with the largest gains observed when only two or three domains are available. Narrow confidence intervals further confirm the stability of our results. By more effectively leveraging domain flow and feature transfer SWAT delivers superior adaptation across diverse data modalities.

### A.3    RESULTS OF OUR METHOD

We present a comparison of our proposed SWAT method with multiple datasets, including Rotated MNIST, Color-Shift MNIST, Portraits, and Cover Type, as detailed in Tables 6 through 9. Each

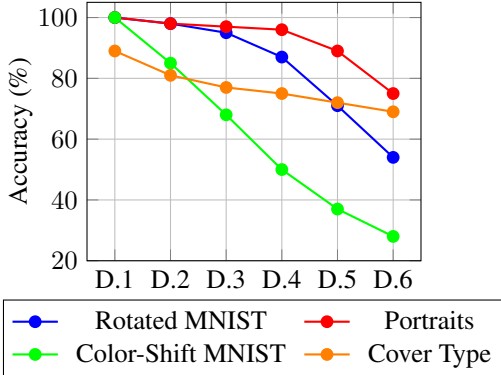

Figure 7: Accuracy of classifiers trained on Domain 1 and evaluated across progressively changing domains (D.2 to D.6) for four datasets: Rotated MNIST, Portraits, Color-Shift MNIST, and Cover Type. The figure illustrates a gradual decrease in accuracy as the domain shift increases, highlighting the impact of domain adaptation challenges.

Table 5: Comparison of SWAT and other GDA methods on 4 datasets given different numbers of intermediate domains.

| Given | Rotated MNIST | | | Given | Color-Shift MNIST | | |
|---|---|---|---|---|---|---|---|
| Domains | GST | GOAT | SWAT | Domains | GST | GOAT | SWAT |
| 2 | $54.9 \pm 0.2$ | $53.5 \pm 1.0$ | $\mathbf{88.1 \pm 1.5}$ | 2 | $27.0 \pm 0.3$ | $72.0 \pm 6.0$ | $\mathbf{98.8 \pm 0.3}$ |
| 3 | $60.0 \pm 0.3$ | $57.2 \pm 0.3$ | $\mathbf{96.1 \pm 0.1}$ | 3 | $34.2 \pm 1.7$ | $83.4 \pm 2.9$ | $\mathbf{99.5 \pm 0.0}$ |
| 4 | $67.2 \pm 0.6$ | $68.4 \pm 1.4$ | $\mathbf{96.4 \pm 0.0}$ | 4 | $55.0 \pm 1.9$ | $89.1 \pm 3.6$ | $\mathbf{99.6 \pm 0.0}$ |
| 5 | $71.9 \pm 0.8$ | $78.8 \pm 0.8$ | $\mathbf{96.5 \pm 0.2}$ | 5 | $66.8 \pm 2.2$ | $94.9 \pm 1.0$ | $\mathbf{99.6 \pm 0.0}$ |
| 6 | $75.6 \pm 1.4$ | $85.8 \pm 0.9$ | $\mathbf{96.7 \pm 0.1}$ | 6 | $74.0 \pm 3.4$ | $95.7 \pm 0.3$ | $\mathbf{99.6 \pm 0.0}$ |

| Given | Portraits | | | Given | Cover Type | | |
|---|---|---|---|---|---|---|---|
| Domains | GST | GOAT | SWAT | Domains | GST | GOAT | SWAT |
| 2 | $75.0 \pm 1.7$ | $78.6 \pm 2.2$ | $\mathbf{85.3 \pm 0.1}$ | 2 | $69.1 \pm 0.1$ | $69.0 \pm 0.0$ | $\mathbf{75.0 \pm 0.0}$ |
| 3 | $75.1 \pm 1.0$ | $80.2 \pm 1.3$ | $\mathbf{84.8 \pm 1.0}$ | 3 | $71.1 \pm 0.2$ | $69.0 \pm 0.0$ | $\mathbf{74.3 \pm 0.2}$ |
| 4 | $78.4 \pm 0.9$ | $80.5 \pm 1.3$ | $\mathbf{86.1 \pm 0.3}$ | 4 | $72.4 \pm 0.1$ | $69.0 \pm 0.0$ | $\mathbf{74.6 \pm 0.1}$ |
| 5 | $76.4 \pm 1.8$ | $79.4 \pm 0.6$ | $\mathbf{87.0 \pm 0.0}$ | 5 | $72.8 \pm 0.1$ | $69.1 \pm 0.1$ | $\mathbf{74.6 \pm 0.1}$ |
| 6 | $80.9 \pm 0.6$ | $83.1 \pm 0.6$ | $\mathbf{87.4 \pm 0.2}$ | 6 | $\mathbf{73.1 \pm 0.1}$ | $69.3 \pm 0.0$ | $73.7 \pm 0.2$ |

experiment was repeated multiple times, with the results shown as mean values along with variance intervals. The leftmost column of each table represents the performance obtained using only adversarial training, which corresponds to the method without flow matching.

In Tables 6 to 9, the column "# Given Domains" indicates the number of domains included in the experiment, comprising both the source and the target domains. The "Inter-domain counts in SWAT" columns indicate the number of inter-domain steps taken between the given domains in the dataset. The entire process is equivalent to including ("# Given Domains - 1") × ("# Inter-domain counts in SWAT + 1") + 1 training step, which includes self-training of GAN and the encoder $f$ and classifier $g$. For example, with four domains and three intermediate steps, the total number of training steps is calculated as (4 - 1) × (3 + 1) + 1 = 13 small steps.

Our results demonstrate the effectiveness of the SWAT method across multiple datasets: Rotated MNIST, Color-Shift MNIST, Portraits, and Cover Type. In each case, we vary the number of given domains and the inter-domain steps in SWAT, comparing the model's performance as the number of inter-domain steps increases.

In the results presented in Table 6 (Rotated MNIST), Table 7 (Color-Shift MNIST), and Table 8 (Portraits), SWAT shows a consistent improvement in accuracy as the number of inter-domain steps increases. Specifically, in Table 6, for the scenario where only the source and destination domains are provided (the first row), the accuracy begins at 83.3% with zero inter-domain steps and progressively

Table 6: Comparison of the accuracy of our method for different given intermediate domains (including source and target domains) on the **Rotated MNIST** dataset, as well as the 68% confidence interval of the mean across 5 runs.

| # Given Domains | # Inter-domain counts in SWAT | | | | |
|---|---|---|---|---|---|
|  | 0 | 1 | 2 | 3 | 4 |
| 2 | $83.3 \pm 0.9$ | $85.0 \pm 0.5$ | $86.1 \pm 0.4$ | $86.9 \pm 0.2$ | $\mathbf{88.1 \pm 1.5}$ |
| 3 | $94.7 \pm 0.5$ | $95.1 \pm 0.7$ | $\mathbf{96.1 \pm 0.1}$ | $\mathbf{96.1 \pm 0.1}$ | $96.1 \pm 0.2$ |
| 4 | $95.6 \pm 0.1$ | $96.3 \pm 0.0$ | $\mathbf{96.4 \pm 0.0}$ | $96.2 \pm 0.1$ | $96.3 \pm 0.0$ |
| 5 | $95.9 \pm 0.1$ | $96.1 \pm 0.1$ | $96.1 \pm 0.2$ | $\mathbf{96.5 \pm 0.2}$ | $\mathbf{96.5 \pm 0.2}$ |
| 6 | $95.9 \pm 0.3$ | $96.4 \pm 0.2$ | $95.5 \pm 1.5$ | $96.6 \pm 0.1$ | $\mathbf{96.7 \pm 0.1}$ |

Table 7: Comparison of the accuracy of our method for different given intermediate domains (including source and target domains) on the **Color-Shift MNIST** dataset, as well as the 68% confidence interval of the mean across 5 runs.

| # Given Domains | # Inter-domain counts in SWAT | | | | |
|---|---|---|---|---|---|
|  | 0 | 1 | 2 | 3 | 4 |
| 2 | $96.9 \pm 0.6$ | $96.6 \pm 1.9$ | $94.9 \pm 5.3$ | $\mathbf{98.8 \pm 0.3}$ | $98.0 \pm 1.0$ |
| 3 | $97.9 \pm 1.9$ | $99.4 \pm 0.1$ | $99.4 \pm 0.0$ | $99.2 \pm 0.4$ | $\mathbf{99.5 \pm 0.0}$ |
| 4 | $99.4 \pm 0.0$ | $\mathbf{99.6 \pm 0.0}$ | $99.5 \pm 0.0$ | $99.5 \pm 0.1$ | $\mathbf{99.6 \pm 0.0}$ |
| 5 | $99.5 \pm 0.0$ | $\mathbf{99.6 \pm 0.0}$ | $99.5 \pm 0.1$ | $99.4 \pm 0.3$ | $99.5 \pm 0.1$ |
| 6 | $\mathbf{99.6 \pm 0.0}$ | $99.4 \pm 0.3$ | $99.2 \pm 0.5$ | $99.4 \pm 0.1$ | $99.5 \pm 0.1$ |

Table 8: Comparison of the accuracy of our method for different given intermediate domains (including source and target domains) on the **Portraits** dataset, as well as the 68% confidence interval of the mean across 5 runs.

| # Given Domains | # Inter-domain counts in SWAT | | | | |
|---|---|---|---|---|---|
|  | 0 | 1 | 2 | 3 | 4 |
| 2 | $82.9 \pm 1.2$ | $84.6 \pm 0.2$ | $85.0 \pm 0.9$ | $85.1 \pm 0.2$ | $\mathbf{85.3 \pm 0.1}$ |
| 3 | $84.3 \pm 0.1$ | $84.3 \pm 0.1$ | $84.7 \pm 0.3$ | $\mathbf{84.8 \pm 1.0}$ | $84.5 \pm 0.1$ |
| 4 | $84.4 \pm 0.6$ | $84.1 \pm 0.1$ | $84.5 \pm 1.8$ | $\mathbf{86.1 \pm 0.3}$ | $85.6 \pm 1.1$ |
| 5 | $86.1 \pm 0.1$ | $\mathbf{87.0 \pm 0.4}$ | $\mathbf{87.0 \pm 0.2}$ | $86.7 \pm 0.3$ | $86.5 \pm 0.9$ |
| 6 | $\mathbf{87.4 \pm 0.2}$ | $87.2 \pm 0.4$ | $86.8 \pm 0.7$ | $86.1 \pm 0.5$ | $86.1 \pm 0.6$ |

Table 9: Comparison of the accuracy of our method for different given intermediate domains (including source and target domains) on the **Cover Type** dataset, as well as the 68% confidence interval of the mean across 5 runs.

| # Given Domains | # Inter-domain counts in SWAT | | | | |
|---|---|---|---|---|---|
|  | 0 | 1 | 2 | 3 | 4 |
| 2 | $74.1 \pm 0.0$ | $\mathbf{75.0 \pm 0.0}$ | $\mathbf{75.0 \pm 0.0}$ | $\mathbf{75.0 \pm 0.0}$ | $\mathbf{75.0 \pm 0.0}$ |
| 3 | $74.2 \pm 0.1$ | $\mathbf{74.3 \pm 0.3}$ | $74.2 \pm 0.5$ | $74.0 \pm 0.1$ | $\mathbf{74.3 \pm 0.2}$ |
| 4 | $74.5 \pm 0.1$ | $\mathbf{74.6 \pm 0.1}$ | $74.5 \pm 0.2$ | $74.3 \pm 0.1$ | $74.3 \pm 0.2$ |
| 5 | $\mathbf{74.6 \pm 0.1}$ | $74.3 \pm 0.7$ | $74.1 \pm 0.3$ | $74.3 \pm 0.2$ | $74.4 \pm 0.1$ |
| 6 | $73.6 \pm 0.3$ | $\mathbf{73.7 \pm 0.2}$ | $\mathbf{73.7 \pm 0.2}$ | $73.5 \pm 0.5$ | $73.5 \pm 0.3$ |

increases, reaching 88.1% at four inter-domain steps. This steady enhancement in performance underscores the value of the additional inter-domain steps in improving SWAT's generalization capacity.

Furthermore, focusing on the scenario with zero inter-domain steps, the results suggest that SWAT continues to exhibit improvements across more complex datasets. This suggests that even without inter-domain steps, the model benefits from the progressive adversarial feature matching, enhancing its ability to adapt and generalize effectively across domains.

In the results presented in Table 9 (Cover Type), SWAT shows relatively stable accuracy across different numbers of inter-domain steps. Unlike other datasets like Rotated MNIST, where accuracy increases noticeably with inter-domain steps, the accuracy on the Cover Type dataset remains relatively stable. This suggests that SWAT may already be achieving near optimal performance with fewer inter-domain steps on this particular dataset. This could suggest that the model has already captured the most critical features of the dataset, or that Cover Type may be less complex compared to the other datasets, requiring fewer inter-domain steps for effective transfer learning.

It is important to highlight that the highest accuracy points are typically found in the upper-right and lower-left corners of the table. This suggests that as the number of given domains increases, the SWAT tends to become more complete, eliminating the need for additional intermediate steps to refine the flow. This observation demonstrates that our method of constructing flows matching between domains is particularly effective when only a few domains are given, and the sliding window adversarial training is highly effective all the time.

## B    COMPUTATIONAL COST COMPARISON

Table 10: Running time and peak GPU memory usage (on an RTX 4090) for SWAT, GOAT, and GST (0 inter-domain step) across four benchmarks with 4 given domains and 2 inter-domain step counts.

| Method | Rotated MNIST | Color MNIST | Portraits | CoverType |
|---|---|---|---|---|
| SWAT | 4 min 56 s / 4488 MB | 4 min 59 s / 4488 MB | 13 s / 5636 MB | 4 min 22 s / 660 MB |
| GOAT | 5 min 38 s / 1886 MB | 5 min 43 s / 1392 MB | 20 s / 1632 MB | 1 min 51 s / 586 MB |
| GST (0-step) | 59 s / 1884 MB | 1 min 1 s / 1884 MB | 4 s / 990 MB | 31 s / 612 MB |

Table 10 compares the running time and peak GPU memory consumption of SWAT against GOAT and GST (with zero inter-domain adaptation steps) on an RTX 4090. SWAT incurs only a modest overhead, due to its additional generators and discriminators, while delivering superior task performance. Notably, SWAT matches or outperforms GOAT in both speed and memory usage on most datasets (e.g., 4 min 56 s/4488 MB vs. 5 min 38 s/1886 MB on Rotated MNIST), demonstrating its practical feasibility for large-scale domain adaptation.

## C    ABLATION STUDY ON LEAST CONFIDENCE

In our experiments, we observed that increasing the rejection rate of low-confidence samples, as discussed in section A.3, improves model accuracy by preventing learning from incorrect samples like Fig. 8. However, excessive rejection can harm the model's generalization ability. This finding is intended to inspire further research in this area.

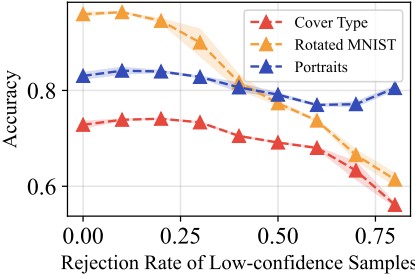

Figure 8: Accuracy vs. Rejection Rate of Low-confidence Samples for Rotated MNIST, Portrait and Cover Type Datasets. Explanation This case involves four given fields and a two-step iteration process is performed between the fields.

## LLM USAGE

**Model and access.** We used OpenAI ChatGPT between July–September 2025.

**Permitted roles.**

- *Writing/editing:* grammar and clarity passes; occasional rephrasing; tightening abstracts/captions.

- *Structural support:* converting bullet notes into section outlines; checklist generation for reporting standards.

- *LaTeX assistance:* resolving formatting issues (tables/figures/macros) and minor template boilerplate.

- *Utility code (non-novel):* small helpers such as CLI parsers, logging stubs, and plotting scaffolds used only for figure generation.

**Explicitly excluded roles.** The LLM was *not* used for problem ideation, novelty claims, algorithm or loss design, theoretical derivations/proofs, dataset construction or labeling, experiment design, hyperparameter search, result selection, or writing any part that constitutes intellectual contributions.

**Verification and safeguards.** All LLM outputs were reviewed and either rewritten or validated by the authors; any code was tested and aligned with our described methodology; citations, equations, and proofs were authored and checked by us; and we avoided introducing unverifiable facts or proprietary content.

