# OpenReview forum: "SWAT: Sliding Window Adversarial Training for Gradual Domain Adaptation"
_ICLR.cc/2026/Conference — ICLR 2026 Conference Withdrawn Submission_

### Official Review · Reviewer_2zyw · 2025-10-14

**Soundness:** 3
**Presentation:** 4
**Contribution:** 2
**Rating:** 2
**Confidence:** 4

**Summary:**

This paper presents Sliding Window Adversarial Training (SWAT), a new framework for Gradual Domain Adaptation (GDA). Instead of aligning source and target domains globally, SWAT introduces a sliding window mechanism that performs localized adversarial training between adjacent intermediate domains. As the window moves gradually from the source to the target domain, the model progressively narrows inter-domain discrepancies, enabling smoother and more stable adaptation. The authors formalize SWAT using Wasserstein GAN loss and cycle-consistency regularization, and evaluate it on six benchmarks (Rotated MNIST, Color-Shift MNIST, Portraits, CoverType, CIFAR-10C, and CIFAR-100C), showing consistent improvements over prior UDA and GDA methods.

**Strengths:**

1. Comprehensive experimental validation on six benchmarks, with consistent performance gains and strong ablations.

2. Clear conceptual motivation — the local-to-global adaptation perspective is easy to understand and practically effective.

3. Good empirical robustness under different corruption levels (CIFAR-10C/100C), suggesting the method’s general applicability.

4. Well-written and reproducible — detailed experimental setup, clear algorithmic description, and open-sourcing commitment.

**Weaknesses:**

1. The sliding-window idea overlaps strongly with existing notions of progressive, incremental, or curriculum-based domain alignment. The adversarial training procedure is a direct adaptation of standard WGAN.

2. No analysis of stability, convergence, or relation to domain discrepancy bounds (e.g., HΔH or Wasserstein continuity).

3. The paper reads more like an improved implementation of GDA rather than an algorithmic innovation for representation learning.

4. Nearly all benchmarks are synthetic or small-scale; it remains unclear whether SWAT scales to more realistic visual adaptation tasks (e.g., GTA→Cityscapes, Office-Home).

**Questions:**

1. How does SWAT differ fundamentally from curriculum domain adaptation (CDA) and progressive self-training (Kumar et al., 2020)?

2. Could the sliding-window schedule be replaced by any continuous interpolation schedule without adversarial training?

3. How is the parameter p chosen or adapted — does it require hyperparameter tuning?

4. What happens when intermediate domains are unavailable or unindexed (e.g., unlabeled target streams)?

5. Have you compared the method against diffusion-based or optimal-transport domain adaptation frameworks beyond WGAN?

---

### Official Review · Reviewer_NndT · 2025-10-31

**Soundness:** 3
**Presentation:** 3
**Contribution:** 2
**Rating:** 2
**Confidence:** 4

**Summary:**

The paper proposes SWAT (Sliding Window Adversarial Training) for Gradual Domain Adaptation (GDA).  In SWAT, the authors build a continuous feature flow of intermediate domains between source and target, and use a sliding window over adjacent domains, which is aligned locally within that window. They train adversarially in both directions using WGAN loss, with cycle consistency and self-training. Experiments are conducted on MNIST and  CIFAR to verify effectiveness.

**Strengths:**

- Decomposes a hard domain shift into many tiny “micro-transfers,” which reduces instability and negative transfer.
- The sliding window only matches nearby domains, not global distributions, which makes adversarial training smoother and easier to optimize.
- The bidirectional adversarial training with consistency regularity is reasonable.
- Experimental results on the included datasets seem to be good.

**Weaknesses:**

- Yet the sliding window is a clever idea, the utilized techniques are common in the DA community. Specifically, the adversarial training with WGAN loss and cycle-consistency loss is widely investigated in DA and cycle-GAN. Thus, I feel that there are few challenges in GDA with the sliding window, and the contribution is limited.
- SWAT trains generators, discriminators, and classifiers jointly over many sliding windows with a curriculum over `p`.  This is heavier than standard self-training or vanilla DANN-style adaptation, and may be harder to reproduce in large-scale settings.
- The included datasets are somewhat too simple. Including more large-scale and modern benchmarks, e.g., ArXiv [1] where the domains continuously change, would significantly strengthen the paper.

[1] Wild-time: A benchmark of in-the-wild distribution shift over time.

**Questions:**

- Since the large language-vision models show strong generalization ability for extracting features. Could the authors explain how the proposed method works/helps at this stage with large language models? Or, where should we use the GDA approaches in practical applications when we have language-vision models?
- SWAT assumes there is a smooth progression of domains from source to target.  Is this progression given (e.g. rotation angles, timestamps), or can it be discovered automatically when you only observe source and target?  In a real scenario where you only have two distributions (source and target) with no obvious gradual steps, how would SWAT be applied?
- The method aligns only within a local sliding window of adjacent domains.  If the window is too short, small errors may accumulate; if it’s too wide, the problem becomes as hard as global alignment again.  How sensitive is performance to the window length (the choice of `p`)? Do you have failure cases or performance curves as `p` changes?
-   SWAT still relies on self-training with pseudo-labels on unlabeled intermediate domains.  If pseudo-labels are poor at some stage, does that error get propagated and amplified to later domains?  Did you run an ablation where you intentionally corrupt pseudo-labels in an intermediate domain and measure how much the downstream domains degrade?
- The authors assume that the distribution shifts between consecutive domains are limited, known as bounded successive divergence. Can you discuss how to identify if the bounded successive divergence holds or not on real data?

---

### Official Review · Reviewer_5hn2 · 2025-10-31

**Soundness:** 3
**Presentation:** 2
**Contribution:** 3
**Rating:** 6
**Confidence:** 4

**Summary:**

Traditional Gradual Domain Adaptation (GDA) transfers knowledge through multiple intermediate domains, yet existing adversarial training methods struggle to maintain stable alignment across these continuous transitions. To address this limitation, the paper proposes Sliding Window Adversarial Training (SWAT), a novel framework that constructs an adversarial feature stream to progressively bridge adjacent intermediate domains. By employing a sliding window paradigm, SWAT narrows the distribution gaps between neighboring domains, enabling smooth, fine-grained, and stable adaptation along the entire domain trajectory. Extensive experiments on six GDA benchmarks demonstrate its effectiveness, thereby validating SWAT’s superior performance and robustness over existing approaches.

**Strengths:**

- First of all, the paper proposes Gradual Domain Adaptation by introducing a continuous domain flow formulation, bridging the gap between static UDA and dynamic adaptation through a hypothesis based on smooth, feature-space probability transition. Technically, the sliding window paradigm enables localized and incremental adaptation between neighboring domains, ensuring stable alignment and preventing abrupt feature shifts.
- The overall framework is well designed and implemented according to the authors’ intent. In particular, the integration of adversarial, cycle-consistency, and self-training losses is conceptually sound and effectively contributes to both stability and semantic preservation throughout the adaptation process.
- In extensive comparison, the authors demonstrate consistent performance gains and robustness under both gradual and high-severity domain shifts.

**Weaknesses:**

- One concern is that the proposed method assumes monotonic and smooth domain sequences, which may limit its applicability to non-linear or multi-factor domain shifts. If the relationships among domains are non-linear or influenced by multiple independent factors, further validation is needed to determine whether the proposed assumptions remain valid and whether the method can still operate effectively under such complex domain transition scenarios.
- Although the paper shows few grammatical errors, its overall organization and logical flow could be improved. For instance, there are structural inconsistencies such as duplicated references (e.g., “Yaroslav Ganin and Victor Lempitsky. Unsupervised domain adaptation by backpropagation. In *International Conference on Machine Learning*, pp. 1180–1189. PMLR, 2015a” appears twice), which should be carefully revised to enhance the paper’s completeness.

**Questions:**

- How does SWAT perform under non-monotonic or multi-dimensional domain trajectories?
- Could SWAT be extended to continual or online adaptation without access to previous domains?

---

### Official Review · Reviewer_vCun · 2025-11-01

**Soundness:** 2
**Presentation:** 1
**Contribution:** 2
**Rating:** 2
**Confidence:** 4

**Summary:**

This paper proposes Sliding Window Adversarial Training (SWAT) to address the problem of Gradual Domain Adaptation (GDA). SWAT constructs a bidirectional adversarial flow and utilizes a sliding window paradigm to break down the large source-to-target domain shift into a sequence of small, manageable alignment steps between adjacent intermediate domains. Experiments demonstrate SWAT's effectiveness across six GDA benchmarks.

**Strengths:**

* This paper proposes an effective method on GDA, and the ablation studies validate the effectiveness of each part of the method.

**Weaknesses:**

* This paper has limited innovations to the prior work. There is prior work [1] with a very similar sliding window idea to this paper. In lines 48-49, you claim that the sliding window mechanism is a new training paradigm in GDA. But the work of [1] also uses a sliding window and a similar parameter $\rho$ to shift from source to target. The authors didn’t compare the difference and the innovation of this work.
* Many statements in the paper are not clear. Lines 42-46 make no sense to me. What do you mean by “a steep gradient will cause discontinuities and unsmooth problems in the manifold space”? I didn’t see any relationship between these two. And in the last sentence of this paragraph, what do you mean by “continuous and smooth manifold hypothesis avoids abrupt changes in decision boundaries”? The motivation of your paper is not clear to me. Some sentences in your related work are also confusing and do not highlight the significance of your work.
* The three advantages you mentioned in Lines 49–71 all stem from the design of the gradual domain adaptation setting itself, which has multiple intermediate domains bridging the source and target domain. It is an overstatement that your method has those advantages.
* The authors should put more effort on the details. It is obvious that the authors don’t put much effort into the writing and notations, because there are so many inconsistencies and mistakes in your math formulas. In Eq. (3), the $D_t$ should be $D_{t+1}$ since you use the model trained on the t-th iteration to assign pseudo-labels for the t+1-th data. In Eq. (2), D_t is defined as a joint dataset of x and y, while in Eq. (3) D_t becomes a marginal dataset of x. You could define a new notation such as D_t^x to represent the marginal one. Besides, since D_t is an empirical dataset, I suggest that the authors use \sum_x over D_t instead of expectation, which is often used for distribution. What are the definitions of $p_t$ and $h$ in Eq. (4)? You have already defined $h$ as a hypothesis in Line 135. But this $h$ in Eq. (4) seems to have a different meaning? Besides, in Lines 187-188, you have $H^i, f_t, g_t$, which are not used in Eq. (4) at all… And later in Eq. (8), you use $g$ to denote the feature generator, while in Line 188, you denote $g$ as the classifier. There are many other typos; I haven’t listed all of them here. This paper needs to be polished.


[1]: Self-Training with Dynamic Weighting for Robust Gradual Domain Adaptation, https://www.arxiv.org/abs/2510.13864

**Questions:**

N/A

---

### Note · Authors · 2025-11-12

I have read and agree with the venue's withdrawal policy on behalf of myself and my co-authors.